# Stereotactic MR-Guided Adaptive Radiotherapy for Pancreatic Tumors: Updated Results of the Montpellier Prospective Registry Study

**DOI:** 10.3390/cancers15010007

**Published:** 2022-12-20

**Authors:** Karl Bordeau, Morgan Michalet, Aïcha Keskes, Simon Valdenaire, Pierre Debuire, Marie Cantaloube, Morgane Cabaillé, Fabienne Portales, Roxana Draghici, Marc Ychou, Eric Assenat, Thibault Mazard, Emmanuelle Samalin, Ludovic Gauthier, Pierre-Emmanuel Colombo, Sebastien Carrere, François-Régis Souche, Norbert Aillères, Pascal Fenoglietto, David Azria, Olivier Riou

**Affiliations:** 1Montpellier Cancer Institute (ICM), University Federation of Radiation Oncology of Mediterranean Occitanie, University Montpellier, INSERM U1194 IRCM, 34298 Montpellier, France; 2Medical Oncology Department, ICM, Montpellier Cancer Institute, University Montpellier, 34298 Montpellier, France; 3Medical Oncology Department, CHU St. Eloi, 34000 Montpellier, France; 4Biometrics Unit, ICM, Montpellier Cancer Institute, University Montpellier, 34298 Montpellier, France; 5Digestive Surgery Department, ICM, Montpellier Cancer Institute, University Montpellier, 34298 Montpellier, France; 6Surgical Department, CHU St. Eloi, 34000 Montpellier, France

**Keywords:** stereotactic MR-guided adaptive radio therapy (SMART), stereotactic body radiation therapy (SBRT), pancreas cancer, pancreatic tumors, locally advanced pancreatic cancer (LAPC), borderline resectable pancreatic cancers (BRPC)

## Abstract

**Simple Summary:**

While the role of radiation therapy in the management of pancreatic tumors remains controversial, new technological modalities allow for safer and more effective radiotherapy treatments. Stereotactic MR-guided Adaptive RadioTherapy (SMART) is an attractive treatment for pancreatic tumors, taking advantage of this challenging tumor location from the continuous image guidance and target tracking, as well as the daily adaptive process. We report in this prospective registry study the largest series of pancreatic SMART to date. Our study confirms the interest of this technique with a high therapeutic index since it is very well tolerated and gives encouraging results in our selected population. Pancreatic SMART could contribute to the improvement of the management of pancreatic adenocarcinoma, whose prognosis remains poor. Its exact place remains to be confirmed in further studies.

**Abstract:**

Introduction: Stereotactic MR-guided Adaptive RadioTherapy (SMART) is a novel process to treat pancreatic tumors. We present an update of the data from our prospective registry of SMART for pancreatic tumors. Materials and methods: After the establishment of the SMART indication in a multidisciplinary board, we included all patients treated for pancreatic tumors. Primary endpoints were acute and late toxicities. Secondary endpoints were survival outcomes (local control, overall survival, distant metastasis free survival) and dosimetric advantages of adaptive process on targets volumes and OAR. Results: We included seventy consecutive patients in our cohort between October 2019 and April 2022. The prescribed dose was 50 Gy in 5 consecutive fractions. No severe acute SMART related toxicity was noted. Acute and late Grade ≤ 2 gastro intestinal were low. Daily adaptation significantly improved PTV and GTV coverage as well as OAR sparing. With a median follow-up of 10.8 months since SMART completion, the median OS, 6-months OS, and 1-year OS were 20.9 months, 86.7% (95% CI: (75–93%), and 68.6% (95% CI: (53–80%), respectively, from SMART completion. Local control at 6 months, 1 year, and 2 years were, respectively, 96.8 % (95% CI: 88–99%), 86.5 (95% CI: 68–95%), and 80.7% (95% CI: 59–92%). There was no grade > 2 late toxicities. Locally Advanced Pancreatic Cancers (LAPC) and Borderline Resectable Pancreatic Cancers (BRPC) patients (52 patients) had a median OS, 6-months OS, and 1-year OS from SMART completion of 15.2 months, 84.4% (95% CI: (70–92%)), and 60.5% (95% CI: (42–75%)), respectively. The median OS, 1-year OS, and 2-year OS from initiation of induction chemotherapy were 22.3 months, 91% (95% CI: (78–97%)), and 45.8% (95% CI: (27–63%)), respectively. Twenty patients underwent surgical resection (38.7 % of patients with initially LAPC) with negative margins (R0). Conclusion: To our knowledge, this is the largest series of SMART for pancreatic tumors. The treatment was well tolerated with only low-grade toxicities. Long-term OS and LC rates were achieved. SMART achieved high secondary resection rates in LAPC patients.

## 1. Introduction

Pancreatic tumors will become the second leading cause of cancer deaths in the United States by 2030 [1]. Projections appear to be similar in France [2]. Adenocarcinoma is the main tumor developing in the pancreas. It has a poor prognosis, with an overall survival rate of about 8% at 5 years, mainly due to its rapid metastatic spread [3,4]. Considering the results of the phase III LAP 07 trial for locally advanced pancreatic cancer (LAPC), radiotherapy is subject to controversy owing to better local control, but no benefit in OS, compared to chemotherapy alone (15.2 vs. 16.5 months) [5].

Pancreatic stereotactic body radiotherapy (SBRT) is an attractive technique because it allows for the delivery of a biological equivalent dose at least as large as in chemoradiotherapy (RCT) but in a shorter time. A meta-analysis shows a better tolerance and possibly a higher efficacy than RCT [6]. However, the proximity of organs at risk (OAR) limits the dose prescription with this treatment modality [7,8].

Stereotactic MR-guided Adaptive RadioTherapy (SMART) is the only technique that delivers high-dose pancreatic SBRT to the tumor, while maintaining optimal OAR protection [9,10,11,12,13,14]. We recently published our initial results with this technique on the first 30 patients treated [15].

The objective of this study was to update our results of SMART for pancreatic tumors.

## 2. Methods and Materials

### 2.1. Patient Selection

After the establishment of the SMART indication in a multidisciplinary board and subsequent validation by the physician in consultation, we included all patients treated for pancreatic tumors from October 2019 and April 2022. A secondary technical board (radiation oncologists and physicists) had to verify the confirmation of treatment on SMART. The inclusion criteria were: nonmetastatic LAPC with stable or responsive disease after the induction of chemotherapy, local recurrent unresectable pancreatic adenocarcinoma after previous pancreatic surgery, and metastatic pancreatic adenocarcinoma after complete or near complete response to chemotherapy with residual primary tumor. Pancreatic metastasis from various primary cancers could be treated, if a systemic agent stabilized any extra pancreatic disease. Pathological assessment was mandatory. The contraindications were comprised of ECOG (Eastern Cooperative Oncology Group) > 2, non-MRI compatible pacemakers, age < 18 years, unstable psychiatric diseases, stomach or duodenal invasions on endoscopy, metal objects, and claustrophobia.

This study was registered in the Health Data Hub (registration number: #1802) and was approved by our local research committee COMERE (ICM-ART 2020/01). All patients signed an informed consent form before treatment.

### 2.2. Radiotherapy Planning and Delivery

Our treatment planning, breath-hold procedure, and daily adaptive workflow have been previously described in detail [15]. Briefly, patients underwent contrast enhanced CT simulation directly followed by 0.35T MRI simulation using the MRIdian^®^. Coregistration by contrast enhanced 1.5T MRI was mandatory in order to optimize tumor delineation and target volumes. MRI images were acquired with true fast imaging with steady state-free precession (TRUFISP) sequences (T1/T2 weighted, breath-hold technique (physiologic end-expiration), 17 to 25 s, 1.6 × 1.6 × 3 mm^3^, or 1.5 × 1.5 × 3 mm^3^ resolution, 45 × 45 × 24 to 54 × 47 × 43 cm^3^ maximum fields of view. The end expiration breath-hold procedure was employed for simulation and treatment. Patients were asked to be compliant to learn the breath process. In any case, radiotherapy therapists had to check “online” the reproducibility of patient position helped by continuous sagittal cine-MR guidance.

Primary tumor (GTV T) and, if necessary, pathologic lymph nodes (GTV N) were delineated on CT and MRIdian^®^ simulation images. Any other imaging considered useful by physicians was co-registrated (1.5T MRI, CT at diagnosis). We created planning target volume (PTV) by application of a 3 mm isotropic extension from GTV. An optimization structure (PTV optimized or PTV opt) was created as follows: PTV opt = PTV − (OAR + 5 mm). OAR dose constraints were strictly prioritized and have been previously reported. The Viewray^®^ treatment planning system (TPS) calculated dosimetry thanks to the Monte Carlo algorithm. As previously described, our target volume dose constraints tried to achieve 95% PTV opt coverage within the 95% isodose, 99% GTV coverage with the 95% isodose, with normalization on D50%. Treatment was delivered using a step and shoot IMRT with 6 MV photons in 14 to 28 beams and 55 to 120 segments. Concurrent chemotherapy was not allowed.

All patients underwent daily adaptive treatment as previously described. After rigid registration of the GTV, a propagation of OAR contours on the daily MR image using deformable image registration was undertaken. OAR contours were medically adjusted (especially digestive OAR). An evaluation and adaptation of the initial plan was performed to obtain a dosimetric benefit on PTV coverage and/or on OAR protection. The electron density map (warped from the CT to the MR images) and the skin contour were verified to ensure correct dose recalculation [16]. A structure with good spontaneous contrast on MRIdian acquisition (usually the GTV itself) was tracked on sagittal images obtained by cine MR. The beam was automatically turned off when 5% or more of the structure was outside a 3 mm threshold from its initial position.

### 2.3. Clinical Assessment, Dosimetric Evaluation, and Endpoints

The primary endpoints were acute (<90 days after SMART) and late (from 90 days) toxicities. Common Terminology Criteria for Adverse Events (CTCAE) v5.0 was used to report toxicities. Secondary endpoints included local control (defined as the absence of RECIST progression in the pancreatic tumor), overall survival (defined by death from any cause from the start of chemotherapy or end of SMART), distant metastasis free survival (defined by RECIST distant relapse or death from any cause from the start of chemotherapy or end of SMART), and dosimetric advantages of adaptive process on OAR and targets volumes. Clinical examination, radiological (CT, MRI, or PET/CT), and biological (blood sample with CA19.9) assessment were recorded for each patient at 1 month and then every 3 months. Specialized pathologists performed an evaluation of the response to neoadjuvant treatment in the tumor. A subgroup analysis was made for LAPC and BRPC patients, in order to compare the overall survival between resected and non-resected patients after SMART. The treatment response was defined as required by Response Evaluation Criteria For Solid Tumors v1.1. Follow up began on the first day of SMART treatment until the death or latest news for each patient. To analyze the impact of an adaptive procedure on recorded values of OAR and target volume coverage (GTV, PTV, and PTV opt), adapted fractions were examined in comparison with predicted fraction (initial plan on the daily image).

### 2.4. Statistical Analysis

The number of observations (n) and their frequency (%) were used to describe qualitative variables. The median and range were recorded for quantitative variables from the patient’s baseline characteristics. The average and standard deviation were registered for dosimetric measures.

The median follow up and clinical outcomes (LC, OS, DMFS) were estimated using the Kaplan-Meier method. A comparison of the survival curves between the resected patients and the non-resected patients was performed by the log rank (Mantel-Cox) test with hazard ratio (Mantel-Haenszel) calculation.

For each adapted fraction delivery, the predicted plan and the adapted plan were compared *a posteriori* by a paired Wilcoxon test. The statistical significance was established at *p* < 0.05. Statistical analyses were performed using Stata v16.0 and GraphPad PRISM v9.

## 3. Results

### 3.1. Patient and Treatment Characteristics

Seventy patients were evaluated after SMART completion between October 2019 and April 2022. The median age was 65 years (range, 39–85). Pancreatic adenocarcinomas represented 90% (*n* = 63) of tumors, and among them 82.6% (*n* = 52) were borderline or locally advanced. One patient was classified as a resectable patient but was unfit for surgery, six patients had local relapse after surgery, and four patients presented oligometastatic disease. Sixty-one patients (87.1%) received chemotherapy before SMART, mainly the FOLFIRINOX regimen (50%). Tumors were predominantly localized in the head of the pancreas (52.9%) and measured 30.8 mm. Lymph node involvement on CT/MRI or PET was negative for 55 patients (78.6%). Among non-pancreatic adenocarcinoma, there was one pancreatic neuroendocrine tumor and 6 pancreatic metastases, 5 from the kidney and 1 from a lung tumor. After induction chemotherapy, the median CA 19.9 decreased from 302 UI/mL (range, 19–3000) to 78 UI/mL (range, 11–802) (Table 1).

### 3.2. Initial Treatment Plans

The reference prescribed dose was 50 Gy (range 30–50) in 5 fractions and this regimen was delivered to 65 patients (93%). A dose reduction was applied in 5 patients and explained by distinctive tumor histology, size, or staging. Three patients received a decreased dose of 40 Gy (two patients with borderline adenocarcinoma and one with metastasis from clear cell renal carcinoma), one patient 35 Gy (metastasis from clear cell renal carcinoma), and one patient 30 Gy (pancreatic neuro endocrine tumor) in 5 fractions. The median duration of fractions was 83 min (range, 52–133) including the patient preparation and the process of adaptive radiotherapy (image registration, OAR delineation, plan adaptation, and treatment delivery). The median PTV was 70.3 cm^3^ (range, 3.8–162). Table 2 presents dosimetric data from the initial target volume and OAR.

### 3.3. Dosimetric Benefits of Adaptive Treatments

All fractions (350) were adapted because of a dosimetric benefit obtained either on PTV coverage or on OAR protection. However, complete dosimetry data could not be retrieved for two fractions due to technical reasons and, therefore, the results are presented for 348 fractions.

The average dosimetric data and comparison between predicted and adapted plans are available on Table 3. Tumor coverage was significantly improved thanks to the adaptive procedure. The PTV opt adapted V100% and V95 % were significantly increased by 5.7% (56.6% to 62.3%, *p* < 0.001) and 4.6% (85.8 to 90.4%, *p* < 0.001) compared to the predicted plans. The benefit of adaptation was explicit for digestives OAR, especially the stomach and duodenum. The Figure 1 shows an example of the benefit of adaptation on OAR sparing (duodenum).

### 3.4. Toxicities

No patients presented radio-induced grade > 2 acute toxicities. The most frequent grade 1–2 toxicities were diarrhea (26%), abdominal pain (30%), and nausea (34%).

After surgery, one patient presented a digestive fistula and another presented an abdominal aneurism. Both were related to post-operative complications after head pancreatic surgery. The evolution was favorable after additional surgical management. One grade 4 and one grade 5 sepsis occurred postoperatively with no relation to radiotherapy.

Acute grade 3 angiocholitis due to tumor compression following biliary prosthesis migration was noted in one patient at their 3 months follow-up and was resolved after changing the biliary prosthesis.

The median follow-up since the end of SMART was 10.8 months for the whole cohort (95% CI: 8.2–13.9). Twenty patients had a follow up less than 6 months after and could not be assessed for late toxicity. Fifty patients were, therefore, assessed for late toxicity. The most common late grade 1–2 toxicities were abdominal pain (46%), diarrhea (40%), and nausea and vomiting (18%) often related to metastatic progression and subsequent chemotherapy treatments. A possibly radio-induced congestive grade 3 duodenal stenosis of the genu superior and proximal D2 was found in a patient 15 weeks after SMART by endoscopic procedure for endoscopic retrograde cholangiopancreatography (ERCP). The Dmax delivered to the duodenum was 30.73 Gy and the V18 Gy was 7.58 cc. It was revealed that ERCP was performed for this patient for a grade 3 angiocholitis originating from the common hepatic duct caused by a biliary prosthesis obstruction, far from the radiotherapy volume and, therefore, was unrelated to SMART. More details are available on Table 4.

### 3.5. Survival Analysis

#### 3.5.1. Whole Cohort (70 Patients)

The median overall survival (OS) from SMART completion was 20.9 months. The 6-months, 1-year, and 2-year OS from SMART completion were, respectively, 86.7% (95% CI: (75.1–93.2%), 68.6 % (95% CI: (53.2–79.9)), and 37.7% (95% CI: (17.4–58.1) (Figure 2A).

Local control from SMART completion at 6 months, 1 year, and 2 year was, respectively, 96.8 % (95% CI: 87.8–99.2%), 86.5% (95% CI: (68.3–94.6)), and 80.7% (95% CI: 58.8–91.7%) (Figure 2B). Among 6 local relapses (8.6%), 4 were located on field edge and 2 inside the field.

#### 3.5.2. LAPC and BRPC Patients (52 Patients)

The median overall survival (OS) was 15.2 months from SMART completion. The 6-months, 1-year, and 2-year OS from SMART completion were, respectively, 84.4% (95% CI: (69.8–92.3)), 60.5% (95% CI: (42.4–74.5)), and 36.3% (95% CI: (15.7–57.4)) (Figure 3A). The median distant metastasis-free survival (DMFS) from SMART completion was 8.3 months. The 6-months and 1-year DMFS from SMART completion were, respectively, 65.4% (95% CI: (50.3–77)) and 33.5% (95% CI: (18.7–49)) (Figure 3B).

The median, 1-year, and 2-year OS from the initiation of induction chemotherapy were 22.3 months 91% (95% CI: (77.7–96.5)) and 45.8% (95% CI: (26.6–63.0)), respectively (Figure 3C).

The median DMFS and 1-year DMFS from the initiation of induction chemotherapy were 14.3 months and 67.5% (95% CI: (51.9–79.1%), respectively (Figure 3D).

The median serum CA 19.9 initially decreased with a nadir at 6 months (52 UI/mL range, 5.2–692) and increased at 1 year (187 UI/mL range, 9–8201).

In the subgroup of borderline and locally advanced pancreatic adenocarcinoma, some patients were discussed for tumor surgery in a multidisciplinary board. These selected patients had to be clinically, radiologically (according to RECIST criteria), and biologically (CA 19.9 decrease) responsive to the treatment. In our cohort, 20 patients (38.5%) were resected including 19 LAPC and 1 BRPC. The median duration between the end of SMART and surgery was 3 months (range 1.1–8.6). All patients underwent complete surgery with negative margins (R0) and the combined pathologic effect of chemotherapy and SMART was estimated at 77.5% (range, 10–100%). One patient experienced a pathologic complete response. Patients were mainly classified ypT2N0 after surgery (40%). Among them, five patients died, including three patients due to a metastatic relapse, one from immediate post-operative complications, and one from undercurrent disease. Seven patients experienced metastatic relapse and one patient both local and metastatic relapses. Resected patients had a median OS from the end of SMART of 21.6 months compared to 11.8 months for unresected patients (*p* = 0.11) (Figure 4).

## 4. Discussion

The management of pancreatic tumors is complex and multidisciplinary, involving several specialties including digestive surgeons, medical oncologists, and radiation oncologists. The prognosis of these tumors is poor and complete resection surgery is the only potentially curative treatment for these patients. For tumors classified as locally advanced, surgery should only be discussed for patients who respond favorably to neoadjuvant chemotherapy and radiotherapy, by trained surgeons operating in high-volume centers (the definition of which is not mutually exclusive). It often require complex reconstructions due to the close proximity of these tumors to arterial and venous vessels. For these tumors thought initially to be unresectable, induction chemotherapy with FOLFIRINOX seems to be the treatment of choice [17]. The place of fractionated chemoradiotherapy in this situation remains debated since the results of the GERCOR LAP07 trial, which did not show any survival benefit compared to Gemcitabine-based chemotherapy, despite better local control and a longer treatment-free time [5]. The results of the recently presented CONKO-007 trial were also disappointing, both in terms of the benefit of fractionated chemoradiotherapy and the impact of FOLFIRINOX, since OS stagnates at 15 months in this situation, although more than 80% of patients had received FOLFIRINOX in induction [18]. Radiation resistance of pancreatic adenocarcinoma was one of the main reasons attributed to this failure of conventional dose chemoradiotherapy [19]. The evolution of technologies has allowed the development of pancreatic stereotactic body radiotherapy (SBRT). No randomized phase III study has shown the superiority of SBRT over conventional chemo-radiotherapy in pancreatic cancer, although a benefit of SBRT seemed to be suggested by the recent meta-analysis, showing an improvement in the overall survival (27% vs. 14% at 2 years), and a decrease in grade 3–4 acute toxicities (5.6% vs. 37.7%) [6]. It is in this context that pancreatic Stereotactic MR-guided Adaptive Radiotherapy (SMART) has developed as a technique allowing for dose escalation on the tumor in a safe way to maintain healthy organs protection [11]. The dosimetric interest of the technique for this localization has been demonstrated by several teams including ours [13,15,20,21]. We confirmed once again this dosimetric benefit with data from 348 adapted fractions, which constitutes to date to our knowledge the largest study on this topic. However, pending the upcoming publication of the ViewRay-sponsored U.S. multicenter Phase 2 SMART pancreas study, there are few published clinical results for SMART in pancreatic tumors to date. Table 5 presents the data from the main SMART studies for pancreatic tumors. The largest clinical study of pancreatic SMART involving 62 patients with pancreatic adenocarcinoma was published in 2022 by the Miami team with a median follow-up of 11 months from the start of SMART [14]. Our study of 70 pancreatic tumors, including 63 pancreatic adenocarcinomas, is, therefore, the largest series to date, with a similar median follow-up of 10.8 months since the end of SMART. Our results, both in terms of local control, OS, and radio-induced toxicity were globally similar to previous reported experiences and seem to have indicated a very favorable therapeutic index of pancreatic SMART. An extremely interesting finding common to all studies of SMART for pancreatic tumors was the excellent tolerability of the treatment. The rates of radiation-induced complications of grade > 2 appeared to be much lower than those reported with chemoradiotherapy. For the subgroup of locally advanced adenocarcinoma (49 patients) and borderline adenocarcinoma (3 patients), our median OS of 22.3 months since the start of chemotherapy was similar to the median OS of 23 months since diagnosis reported by the Miami team. These results compared favorably to the median OS of 15 to 16 months obtained in the LAP07 and CONKO-007 studies (Table 6).

One of the remarkable findings of our study was the high rate of pancreatic resection after neoadjuvant treatments of locally advanced tumors considered initially unresectable. Compared with a recent review of the SBRT literature that reported resection rates of 7–18%, 38.5% of patients in our cohort underwent tumor resection (20 patients) [22]. The Miami and Saint Louis teams reported resection rates of 22.6% and 9.1%, respectively [12,14]. In our analysis, all patients were resected with healthy margins and the histological therapeutic effect was 77.5%. In our first publication of the first 30 patients treated with SMART, resected patients benefited in overall survival when compared to non-resected patients [15]. However, updated data on 70 patients showed a trend towards improved overall survival during the first year of follow-up, but not statistically significant (21.6 months compared to 11.8 months). These results were potentially impacted by the early postoperative death of one patient. This aggressive management appeared nevertheless feasible in experienced teams, with the rates of severe postoperative complications being typical in these high-risk surgeries.

Our study presented some limitations. First, our study population was heterogeneous, with seven patients presenting a neuroendocrine tumor or pancreatic metastases of kidney cancer. Similarly to our previous report, we decided to keep these patients for dosimetric and toxicity analysis as the treatment site, anatomical, and dosimetric characteristics were similar, but to perform a subgroup analysis on BRPCs and LAPCs regarding survival data. Moreover, our study remained a single-center experiment with limited follow-up and our results still must to be confirmed by multicenter experiments with longer follow-up. Finally, although the patients came from a prospective registry, the data analysis was performed in an ambispective way, which may represent a constitutive bias. Hence, we recently opened the French multicenter GABRINOX-ART prospective phase II non-randomized trial. It assesses the contribution of alternative intensified FOLFIRINOX and GEMCITABINE ABRAXANE regimen in LAPC patient. In case of stable or responsive disease, the protocol is followed by SMART to the pancreatic tumors. Reliable results from this trial are awaited. Additionally, the final results of the multicenter SMART pancreas study sponsored by Viewray might add significant knowledge on the topic.

## 5. Conclusions

The updated results of our study supported that SMART for pancreatic tumors is a novel and feasible technique. Daily adaptation benefit is confirmed for tumor coverage and OAR sparing. Acute and late gastro intestinal toxicities are low and first oncological outcomes are promising for local control and overall survival. This study achieved the highest published secondary resection rate in LAPC patients.

## Figures and Tables

**Figure 1 cancers-15-00007-f001:**
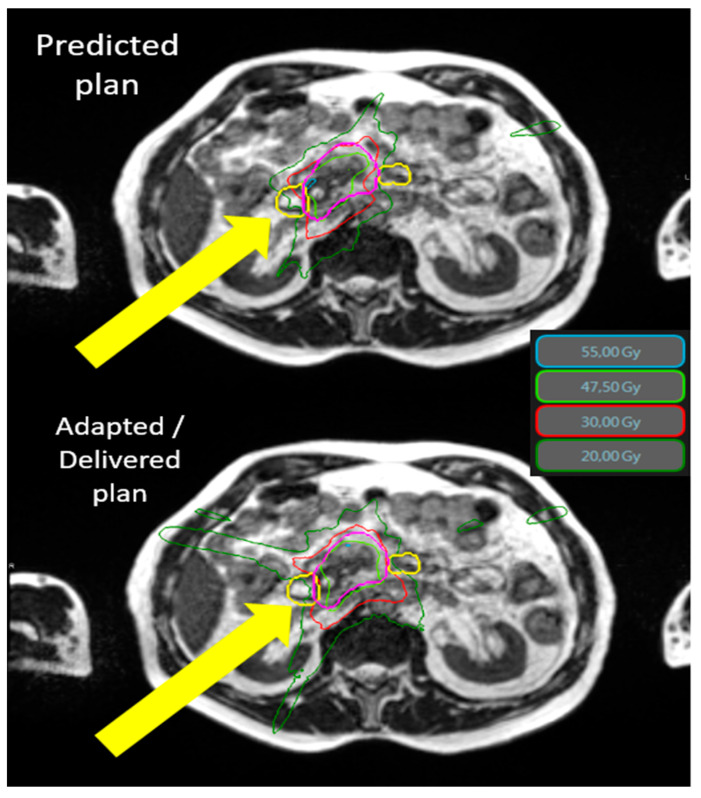
Typical SMART dosimetry showing the predicted and adapted/delivered dosimetry for locally advanced pancreatic cancer. An example of the dosimetric benefit between the predicted plan (**top**) and the adapted plan (**bottom**). The yellow arrows demonstrate the protection of the duodenum (yellow line) by the adaptive process. The 55 Gy (cyan) and 30 Gy (thin red line) isodoses no longer cross the duodenum in the lower image. Isodose 55 Gy in cyan, 47.5 Gy in green, 30 Gy in red, and 20 Gy in dark green. The duodenum is in yellow and PTV is in pink. Abbreviations: Gy = gray 3.4. Toxicities.

**Figure 2 cancers-15-00007-f002:**
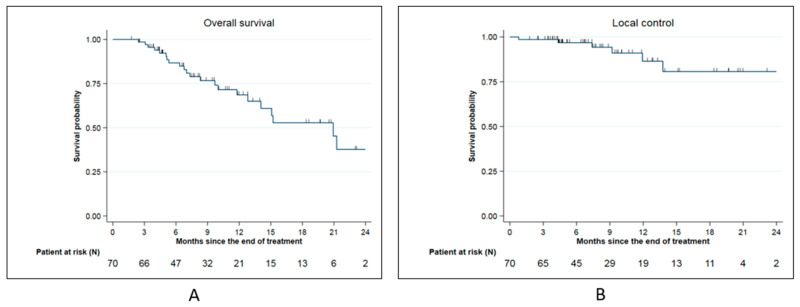
Survival date for the whole cohort: (**A**): Overall survival for the whole cohort; (**B**): Local control for the whole cohort.

**Figure 3 cancers-15-00007-f003:**
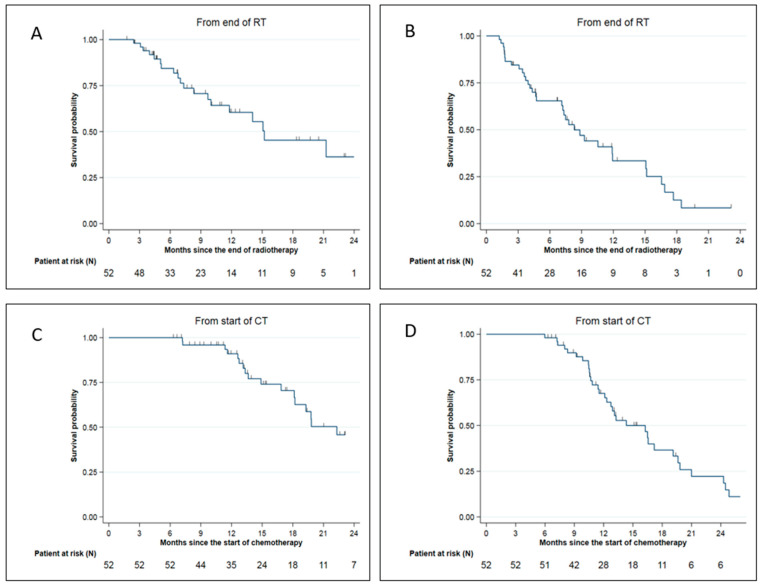
Survival data for LAPC and BRPC patients: (**A**) Overall survival from SMART completion; (**B**) Distant metastasis-free survival from SMART completion; (**C**) Overall survival from chemotherapy start; (**D**) Distant metastasis-free survival from chemotherapy start.

**Figure 4 cancers-15-00007-f004:**
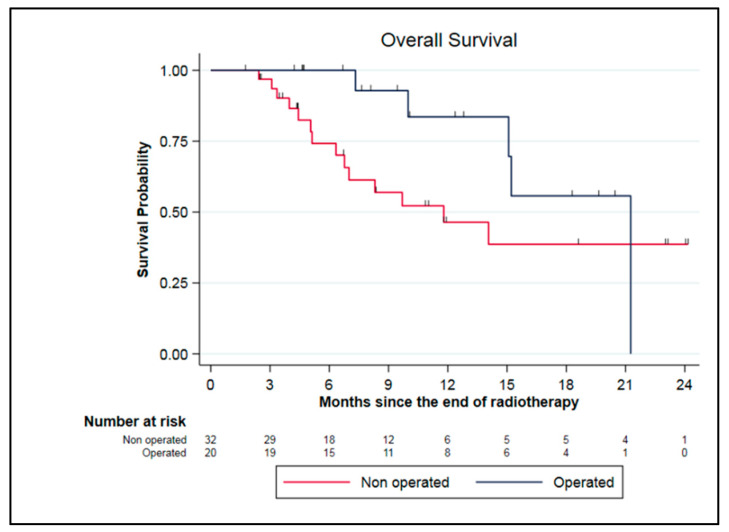
Survival data for LAPC and BRPC patients: comparison between resected and unresected patients.

**Table 1 cancers-15-00007-t001:** Patients baseline characteristics.

**Sex**	
Women	34 (48.6%)
Men	36 (51.4%)
Median age (range)	65 years (39–85)
**Pathology**	
Pancreatic adenocarcinoma (PA)	63 (90%)
Pancreatic neuroendocrine tumor	1 (1.4%)
Metastasis from kidney tumor	6 (8.6%)
**Stage among PA**	
Resectable	1 (1.6%)
Borderline	3 (4.8%)
Locally advanced	49 (77.8%)
Local relapse	6 (9.5%)
Metastatic	3 (4.8%)
Metastatic and local relapse	1 (1.6%)
**Previous Treatment**	
Chemotherapy	56 (80%)
Pancreatic Surgery	3 (4.3%)
Pancreatic Surgery + Chemotherapy	5 (7.1%)
None	6 (8.6%)
**ECOG score**	
0	25 (35.7%)
1	42 (60%)
2	3 (4.3%)
**Chemotherapy Regimen for PA**	
FOLFIRINOX	35 (55.6%)
GEMCITABINE-ABRAXANE	1 (1.6%)
FOLFOX	6 (9.5%)
GEMCITABINE	2 (3.2%)
FOLFIRI	1 (1.6%)
Several protocols	16 (25.4%)
None	9 (14.3%)
**Localization**	
Head	37 (52.9%)
Body/Tail	25 (35.7%)
Mixed location	2 (2.9%)
Local relapse	6 (8.6%)
**Lymph Node Involvment °**	
Yes	15 (21.4%)
No	55 (78.6%)
**Median CA 19.9 before Chemotherapy** (range)	302 UI/mL (19–3000)
**Median CA 19.9 before SMART** (range)	78.1 UI/mL (11–802)
**Average size of Pancreatic Tumor** (standard deviation) (Min–Max)	30.8 mm (9.8) (12–55)

° On CT/MRI/PET.

**Table 2 cancers-15-00007-t002:** Median (min-max) dosimetric data for initial plans.

Total Dose (Gy)	50 (65 Patients)
	40 (3 Patients)
	35 (1 Patient)
	30 (1 Patient)
Total Treatment Duration (days)	7 (5–14)
Fraction Dose (Gy)	10 (6–10)
Median PTV (cm^3^)	70.3 (3.8–162)
Fraction Duration (min)	82.6 (52–133)
**PTV opt**	
V100% (%)	63.2 (37.2–83.1)
V95% (%)	94.3 (68.9–99.9)
V80% (%)	99.9 (92.9–100.0)
D98% (Gy)	44.4 (28.5–48.2)
D95% (Gy)	46.5 (29.1–48.8)
D2% (Gy)	53.0 (32.2–55.2)
**PTV**	
V100% (%)	53.9 (33.8–78.5)
V95% (%)	80.3 (57.5–98.5)
V80% (%)	92.4 (72.7–100.0)
D98% (Gy)	28.5 (12.6–84.5)
D95% (Gy)	35.8 (15.2–48.6)
D2% (Gy)	52.9 (32.1–55.1)
**GTV**	
V100% (%)	66.3 (40.5–93.4)
V95% (%)	91.3 (65.1–100.0)
V80% (%)	97.6 (77.9–100.0)
D98% (Gy)	38.7 (17.8–49.6)
D95% (Gy)	43.8 (22.4–49.9)
D2% (Gy)	53.1 (32.3–55.9)
**Kidney**	
V18 Gy (cm^3^)	4.4 (0–25.1)
**Spinal Cord**	
Dmax (Gy)	17.9 (6.5–24.6)
**Stomach**	
Dmax (Gy)	30.1 (0.9–34.6)
V18 Gy (cm^3^)	9.6 (0.0–30.5)
**Duodenum**	
Dmax (Gy)	30.0 (12.7–33.4)
V18 Gy (cm^3^)	4.3 (0.0–11.4)
**Small Intestine**	
Dmax (Gy)	28.7 (1.5–34.5)
V19.5 Gy (cm^3^)	3.9 (0.0–20.6)
**Large Intestine**	
Dmax (Gy)	27.9 (5.9–33.7)
V25 Gy (cm^3^)	0.2 (0.0–9.0)

**Table 3 cancers-15-00007-t003:** Average target volume and OAR dosimetric results for predicted and adapted plans.

Target Volume	Predicted Plan (Standard Deviation)	Adapted Plan (Standard Deviation)	*p*-Value
**PTV opt**			
V100% (%)	56.6 (17.9)	62.3 (12.6)	**≤0.001**
V95% (%)	85.8 (9.0)	90.4 (8.4)	**≤0.001**
V80% (%)	96.4 (4.1)	98.9 (2.0)	**≤0.001**
D98% (Gy)	37.4 (7.1)	42.7 (4.5)	**≤0.001**
D95% (Gy)	41.7 (5.8)	44.8 (4.1)	**≤0.001**
D2% (Gy)	52.6 (4.1)	52.5 (4.2)	0.003
**PTV**			
V100% (%)	51.6 (17.0)	55.4 (12.6)	**≤0.001**
V95% (%)	78.4 (11.2)	80.8 (11.7)	**≤0.001**
V80% (%)	90.5 (7.5)	91.2 (7.5)	0.003
D98% (Gy)	29.3 (9.3)	29.9 (10.4)	0.163
D95% (Gy)	34.5 (9.2)	35.3 (9.9)	0.021
D2% (Gy)	52.5 (4.0)	52.2 (4.2)	0.001
**GTV**			
V100% (%)	63.2 (19.3)	68.3 (13.4)	**≤0.001**
V95% (%)	88.4 (9.7)	89.8 (9.4)	**≤0.001**
V80% (%)	95.3 (5.4)	95.5 (5.5)	0.171
D98% (Gy)	37.1 (9.6)	37.0 (10.3)	0.677
D95% (Gy)	41.1 (7.9)	41.4 (8.6)	0.152
D2% (Gy)	52.7 (4.1)	52.5 (3.8)	0.003
**OAR**	**Predicted Fraction (SD)**	**Adapted fraction (SD)**	** *p* ** **-value**
**Kidney**			
V18 Gy	5.5 % (6.1%)	5.8 % (5.7%)	0.006
**Spinal Cord**			
Dmax	17.5 Gy (3.7 Gy)	17.6 Gy (3.8 Gy)	0.544
**Stomach**			
V18 Gy	14.6 cm^3^ (11.2 cm^3^)	9.7 cm^3^ (7.4 cm^3^)	**≤0.001**
Dmax	34.3 Gy (11.8 Gy)	27.5 Gy (6.5 Gy)	**≤0.001**
**Duodenum**			
V18 Gy	5.7 cm^3^ (5.2 cm^3^)	4.3 cm^3^ (3.6 cm^3^)	**≤0.001**
Dmax	34.5 Gy (12 Gy)	27.5 Gy (5.9 Gy)	**≤0.001**
**Small Intestine**			
V19.5 Gy	5.6 cm^3^ (8.4 cm^3^)	3.4 cm^3^ (3.9 cm^3^)	**≤0.001**
Dmax	31.8 Gy (11.1 Gy)	26.6 Gy (6.2 Gy)	**≤0.001**
**Large Intestine**			
V25 Gy	1.7 cm^3^ (3.4 cm^3^)	1.1 cm3 (2.2 cm^3^)	**0.017**
Dmax	25.7 Gy (10.3 Gy)	23.9 Gy (7.6 Gy)	**≤0.001**

**Table 4 cancers-15-00007-t004:** SMART-related acute and late toxicities.

CTCAE v5.0	Acute Toxicity (0–90 Days)	Late Toxicity (90 Days–1 Year)
	(70 Patients)	(50 Patients)
Abdominal Pain		
g0	49→(70.0%)	27→(54.0%)
g1	17→(24.3%)	14→(28.0%)
g2	4→(5.7%)	9→(18.0%)
Nausea/Vomiting		
g0	46→(65.7%)	41→(82.0%)
g1	18→(25.7%)	4→(8.0%)
g2	6→(8.6%)	5→(10.0%)
Gastritis/Enteritis		
g0	67→(95.7%)	50→(100.0%)
g1	3→(4.3%)	0
Gastro-Duodenal Ulcer		
g0	70→(100%)	50→(100%)
Post-operative Digestive Fistula (20 patients operated)		
g0	18→(97.1%)	17→(94.0%)
g1	0	0
g2	0	1→(2.0%)
g3	2→(2.9%)	2→(4.0%)
Diarrhea		
g0	52→(74.3%)	30→(60.0%)
g1	15→(21.4%)	11→(22.0%)
g2	3→(4.3%)	7→(14.0%)
g3	0	2→(4.0%)

**Table 5 cancers-15-00007-t005:** Main published SMART studies for pancreatic tumors. (Local control and overall survival since end of SMART). Abbreviations: 1y = 1-year rate; 2y = 2-year rate.

Teams, First Author	Year	Number of SMART Patients	Median Dose (Gy)	Fractions	Folllow-Up (Months)	Local Control	Overall Survival	GI Toxicity > 2
Multicenter, Rudra [11]	2019	16	45–52	5	17	2y = 77%	2y = 49%	Acute = 7%
Saint Louis, Hassanzadeh [12]	2021	44	50	5	16	1y = 83.4%	1y = 68.2%	Acute = 0%
						2y = 59.3%	2y = 37.9%	Late = 4.6%
Miami, Chuong [14]	2022	62	50	5	11	1y = 98.2%	1y = 53.8%	Acute = 4.8%
						2y = 68.8%	2y = 27.7%	Late = 4.8%
Montpellier, Bordeau (current study)	2022	70	50	5	11	1y = 86.5%	1y = 68.6%	Acute = 0%
						2y = 80.7%	2y = 37.7%	Late = 1.4%

**Table 6 cancers-15-00007-t006:** Comparison of studies, including radiotherapy for advanced pancreatic adenocarcinoma: radiotherapy arm of the studies LAP07 and CONKO-007 including fractionated chemoradiotherapy and chemotherapy (above the line) versus studies including SMART after induction chemotherapy (below the line).

Teams, First Author	Year	Number of PA Patients	Median Dose (Gy)	Fractions	Overall Survival *	Progression Free Survival *	Distant Metastasis Free Survival *	Resection Rate	GI Toxicity > 2
Multicenter, Hammel [5]	2016	133	54	30	1y = 65.4% 2y = 19.5% m = 15.2 mths	1y = 33.8% 2y = 6% m = 9.9 mths	N/A	4%	Acute = 5.9%
Multicenter, Fietkau [18]	2022	168	50.4	28	1y = 71.3% 2y = 34.8% m = 15 mths	1y = 56.3% 2y = 24.1% m = N/A	N/A	36.3%	N/A
Miami, Chuong [14]	2022	62	50	5	1y = 90.2% 2y = 45.5% m = 23 mths	1y = 88.4% 2y = 40% m = 20 mths	N/A	22.6%	Acute = 4.8% Late = 4.8%
Montpellier, Bordeau (current study)	2022	63	50	5	1y = 91% 2y = 45.8% m = 22.3 mths	N/A	1y = 67.5% 2y = 22.2% m = 14.3 mths	38.5%	Acute = 0% Late = 1.4%

* Since randomisation for the study of Hammel and Fietkau; since diagnosis for the study of Chuong; since initiation of induction chemotherapy for the current study. Abbreviations: 1y = 1-year rate; 2y = 2-year rate; m = median survival; N/A: data not available.

## Data Availability

Data available upon request.

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
