# Peer review of "Stereotactic MR-Guided Adaptive Radiotherapy for Pancreatic Tumors: Updated Results of the Montpellier Prospective Registry Study"

_cancers, 2022, doi:10.3390/cancers15010007_

Round 1

Reviewer 1 Report

In this manuscript, the authors report a series of 70 patients with pancreatic tumor treated with SMART in one center. Tolerance was excellent with no grade 3-4 acute toxicity and 4% of severe late toxicity. Median overall survival was very encouraging with a high rate of secondary resection for patients with LA tumors. The manuscript is clear and well written. However, it could be improved before being suitable for publication.

Major comment: the series is heterogeneous including 10%of patients with another disease than pancreatic adenocarcinoma, with a very different prognosis. Moreover, most of them did not receive 50 Gy as the others, precluding the conclusions of the dosimetric analysis. These 7 patients should be excluded from the study.

Minor comments:

- in Table 1, 6 patients were treated for a local relapse whereas 8 had previous pancreatic surgery, I do not understand these numbers

- in Table 1, 35 patients among 63 with PA received FOLFIRINOX, i.e. 57% rather than 50%. Please edit the percentages

- in Table 1, please add the extremes for the tumor size

- please explain what is "PTV-opti"

- Table 2 could be suppressed

- Table 3 would be easier to read if D98%, D95%, and D2% were expressed in percentage rather than in Gray

- is the patient who experienced a grade 3 duodenal stenosis the same than the one who had a grade 3 angiocholitis? It is unclear

- In paragraph 3.5.1, median OS was 20.9 months but was it calculated from the start of induction chemotherapy or from the end of SMART? Same question for local control

- please add the extremes for the time between the end of SMART and surgert

Remarks :

- radiation oncologist should be preferred than radiation therapist (technician)

- radiation technician should be replaced by radiation therapist

- radiochemotherapy should be replaced by chemoradiotherapy or chemoradiation

- unresectable should be preferred than inoperable (in the Discussion section)

Reviewer 2 Report

K. Bordeau et al. report an update of their prospective study assessing the benefit of stereotactic MR-guided Adaptive radiohterapy (SMART) for pancreatic cancer patients.

The results are clearly presented and very interesting. I have only some minor concerns:

1) The labeling of Figure 1 is not clear. The isodose line for 55 Gy (yellow) is very faint/almost not visible. The authors should change the color. The colour used to delineate the duodenum is roughly the same than for the 20-Gy isodose, making it very confusing. The arrow pointing at the duodenum should be the same colour than the line, and clearly different from any other colour used to draw isodose lines on the figure.

2) The authors should elaborate on the discussion by including the results of the LAP07 and the CONKO-007 studies in Table 5. A new table providing information such as median survival (OS, DMFS) and resectablility for all 4 SMART studies compared to the LAP07 and the CONKO-007 studies would be a clear improvement.

3) The authors should discuss the differences in resection rates from the different studies, and might formulate recommendations for further studies. 

Round 2

Reviewer 1 Report

Thank you for your answers.